# Spanish Cross-Cultural Validation of the Electronic Version of the International Index of Erectile Function-5 (IIEF-5)

**DOI:** 10.3390/ijerph19053115

**Published:** 2022-03-07

**Authors:** Raquel Lizarraga-Limousin, Esther M. Medrano-Sánchez, Esther Díaz-Mohedo, Lorena Vergara-de-Carlos

**Affiliations:** 1Physical Therapist, Arbol de Guernica St., 20400 Tolosa, Spain; rlizarragalimousin@gmail.com; 2Department of Physical Therapy, University of Seville, Avenzoar St., 41009 Seville, Spain; 3Department of Physical Therapy, University of Málaga, Francisco Peñalosa Av., 29071 Málaga, Spain; estherdiaz@uma.es; 4Physical Therapist, Apalaz St., 31292 Zurukuain, Spain; lorezuru@gmail.com

**Keywords:** erectile dysfunction, sexual dysfunction

## Abstract

The IIEF-5 questionnaire is a validated scale used as a diagnostic tool in erectile dysfunction (ED). This simplified version includes five items that focus on erectile function and satisfaction during sexual intercourse; it has favourable properties for detecting the presence and severity of erectile dysfunction The main objective of this study was to make a cross-cultural adaptation to Spanish of the IIEF-5 scale and to evaluate its psychometric properties of validity, reliability, and feasibility in the Spanish population. Validation of the IIEF-5 included: (i) professional translation of the scale; (ii) scientific evaluation of the translation; (iii) professional retranslation; (iv) assessment of 10 individuals to test correct comprehension and idiomatic adequacy; (v) validation of the IIEF-5 by an online survey. The study sample consisted of 100 participants, who received the online form either directly or through other participants who distributed it. Participants obtained a mean score of 22.3 (SD 2.7), implying normal erectile function. However, 23 results of mild dysfunction (*n* = 23) and 2 of mild to moderate dysfunction (5.1%) were identified. A Cronbach’s alpha coefficient of 0.75 was obtained for the total of the final questionnaire, indicating high reliability. Validity analysis had a value of 0.784 (>0.5) and was therefore considered appropriate. The IIEF-5 scale is a reliable tool to test ED, and its Spanish version is satisfactorily understood by patients.

## 1. Introduction

Erectile dysfunction (ED) is the inability to achieve and/or maintain an erection sufficient for satisfactory sexual performance [1]. According to published studies, the variability of prevalence is very wide, ranging from 10 to 52% in men between 40 and 70 years of age [2]. In Spain, almost 50% of the population over 70 years of age suffers from ED to a greater or lesser extent [3] and it is mainly associated with cardiovascular risks, but also with metabolic diseases and chronic kidney disease [3]. It may also occur secondary to pelvic floor muscle dysfunction (lack of pelvic floor muscle control, reduced activation or strength) as erectile strength may be directly affected by reduced tone and alterations in the contractile patterns of this musculature [4].

Most men with ED have suffered from a significant decrease in their quality of life and may experience fear, anxiety, confusion, embarrassment, and self-doubt, which can have a significant impact on social interaction, leading to personal isolation [5]. It is therefore of great importance to approach these patients from a biopsychosocial point of view, in which the different degrees of affectation of the patients’ quality of life are assessed by means of specific and validated questionnaires, and not only the presence or absence of the different symptoms they may suffer due to ED.

From a conservative point of view, the therapeutic approach to ED may include various therapies, including oral pharmacology (phosphodiesterase 5 inhibitors), testosterone replacement therapy, psychosexual counselling, prostaglandin injections, vacuum pumps [6] and pelvic floor training [4].

Nowadays, it is very difficult for a patient to spontaneously declare that he has ED. It is difficult and unlikely that general practitioners will naturally address issues of an intimate and private nature or initiate a conversation about the individual’s sexual concerns, as they fear intrusion and reluctance to discuss sexual performance, which makes it difficult to address [7]. Specialists, such as cardiologists, see many patients with erectile dysfunction because of the common pathophysiology between coronary health disease and erectile dysfunction. Despite patients’ expectations, sexual-functioning problems remain unaddressed in these consultations [8,9]. A study found that in a questionnaire completed by 3291 men suffering from ED, embarrassment about being able to talk about it naturally was 1 of the most common reasons for not seeking treatment with a PDE-5 inhibitor. In addition, the survey showed that, if a physician initiated a discussion of ED, 80% of men who had not previously sought medical help would be willing to discuss their problem [6].

To improve early detection, it is necessary to use validated instruments [9], and we have found several in the literature; however, not all of them have been validated. 

The “gold standard” ED severity assessment scale is the SHIM scale [10,11], which covers the most relevant domains of male sexual function in 15 items: erectile function (6 items), orgasmic function (2 items), sexual desire (2 items), satisfaction during intercourse (3 items) and overall satisfaction (2 items). It is a self-administered questionnaire whose sensitivity and specificity have been demonstrated in detecting treatment-related changes in patients suffering from ED [11]. Another questionnaire is the Erectile Function Pineda Visual Analog Scale (EFP-VAS), which is a prototype scale for ED patients [12]. This could be used as an alternative to the Sexual Health Inventory for Men, especially in patients with academic and linguistic limitations, as it is a simple scale that is easy to complete and understand; however, it is currently not validated. The Erection Hardness Score (EHS) is also used in patients with ED and measures erectile hardness; it has only 1 item, with a Likert scale from 1 to 4 [13].

There is also a simplified version of the SHIM scale, also called IIEF-5, which has been shown to be valid as a diagnostic tool in ED [14]. This simplified version includes five items that focus on erectile function and satisfaction in sexual intercourse; it has favourable properties for detecting the presence and severity of erectile dysfunction [15]. In addition, it is a validated questionnaire for evaluating and assessing the effectiveness of ED treatment [16], and also as a screening tool in general urological practice [17].

While the SHIM scale is validated in more than 30 languages (including Spanish), its shorter version, the IIEF-5, has not yet been validated in the Spanish population; moreover, according to the latest systematic review, it “requires further research on internal consistency, measurement error and responsiveness; the most urgent issue for future research is to determine the unidimensionality of the scale” [18]. The COVID-19 pandemic situation has challenged us to create new, easy, and safe methods of administration that do not require self-administration by a healthcare professional on-site.

Therefore, this study’s research question was if the IIEF-5 scale could be easily adapted to the Spanish culture, and if it could be administered online. According to this question, and in light of the background found in the literature, the main objective of this study is to perform a cross-cultural electronic adaptation of the IIEF-5 erectile dysfunction scale into Spanish and to evaluate its psychometric properties of validity, reliability and feasibility in the Spanish population.

## 2. Materials and Methods

### 2.1. Study Design, Participants and Procedure

A cross-sectional, observational study was designed to validate the IIEF-5 scale in the Spanish population. To conduct the study, we followed the STROBE guidelines [19] and the guidelines of “The Checklist for Reporting Results of Internet E-Surveys” (CHERRIES) [20].

We followed the suggestions of Wild et al. [21] who strongly recommended using the translation–back translation method with bilingual translators and then performing a thorough analysis of the new version to identify discrepancies and to verify that the questionnaire would be clearly understood by study participants.

The inclusion criteria for selecting participants were as follows: male, native Spanish speakers and of legal age. Exclusion criteria were non-acceptance of informed consent and incorrect or incomplete completion of the survey. A snowball exponential discriminative sampling technique was used.

The Spanish version of the IIEF-5 was administered until a sample of at least 100 participants was reached. The sample size was based on the general recommendations of Terwee et al. [22], who recommend at least 7 subjects the assessment of the measures (with a total sample of 100 individuals or more). As the IIEF-5 scale has only 5 items, the authors estimated that a sample of 100 participants guaranteed a proper psychometric analysis. Nevertheless, a sample size calculation was made using the free software G* Power (G* Power 3.1.9.4. version, Universitat Dusseldorf, Dusseldorf, Germany). For a target power of 0.90, a β of 10%, an accuracy of 3% and an α of 0.5, we needed a sample of 92 participants. Taking for granted that there would be some losses, a number of 100 participants was finally set as an ideal sample size.

The questionnaire was distributed online, via email and WhatsApp. The Google Forms platform was used for the completion and distribution, where, with the prior informed consent of the participant, they had access to the Spanish version of the IIEF-5. The online method was chosen as it is a useful, fast, flexible, and inexpensive way to obtain information [23]. The response rate could not be controlled.

#### 2.1.1. Translation–Back Translation Method to Validate the Questionnaire

Two bilingual Spanish physical therapists (E.M.M.S. and L.V.C.), both of whom were specialists in treating pelvic floor and sexual dysfunctions and who perfectly understood the content and purpose of the questionnaire, collaborated to avoid errors in forward translation. Two initial translations were completed. An agreement on the first version of the questionnaire in Spanish was reached during a meeting between the other researchers and the translators.

#### 2.1.2. Back Translation of the Questionnaire into English

After the review by experts, this first translated version was sent separately to 2 bilingual native English-speaking translators who specialize in research papers in the health sciences and who had not seen the original questionnaire [24]. Once the 2 back translations were completed, a meeting was held with the translators to ensure that the translations did not differ conceptually from the original version. Thus, we were able to validate the first translation into Spanish.

#### 2.1.3. Pilot Study

To identify potentially misleading words or questions and to verify that the scale items would be perfectly understood by the participants, we conducted a pilot study with a preliminary draft and 10 men who matched the inclusion criteria. The participants in this pilot study were undergraduate physiotherapy students and patients from a urology practice. The questionnaire was administered online, in the same way as planned for the final sample. During the implementation of the questionnaire, the authors kept in contact with the volunteers by email to allay their possible doubts and to find out if they had detected any confusing expressions or terms. No changes were reported by the volunteers, so there was no need to make any changes to the Spanish draft of the questionnaire.

#### 2.1.4. Final Version of the IIEF-5

Once the final version adapted into Spanish was obtained, its validation was undertaken by verifying the metric properties of the questionnaire for the entire simple (N_100).

### 2.2. Measurement

The questionnaire was distributed online, via email and WhatsApp. The Google Forms platform was used for the completion and distribution, where with the prior informed consent of the participant, they had access to the Spanish version of the IIEF-5. The online method was chosen because it is a useful, fast, flexible, and cheap way to obtain information [24], as well as being the reference method in this time of pandemic with mobility restrictions. The response rate could not be controlled.

### 2.3. Instruments

The IIEF-5 questionnaire is an abbreviated test of the SHIM questionnaire that allows one to classify the severity of ED in the male population according to the score obtained. It consists of five questions, each of which is scored on a five-point ordinal scale where lower values mean greater severity of ED. Thus, an answer with a value of 0 is considered to be the least functional, while an answer with a value of 5 would be the most functional. The possible scores on this questionnaire range from 1 to 25; a score above 21 is considered normal erectile function and below this value would be considered to a greater or lesser degree depending on the score obtained for ED. Therefore, according to the IIEF-5 scale we can classify it into five different categories: severe (1–7), moderate (8–11), mild to moderate (12–16), mild (17–21), or no ED (24–27) [9,12].

### 2.4. Statistical Analysis

We used SPSS version 26.00 for Windows (SPSS Inc., Chicago, IL, USA) for all of the statistical analyses.

First, for the description of the sample, descriptive mean and standard deviation with maxima and minima (range and 95% confidence interval) were calculated. The score obtained on the scale translated into Spanish was also calculated.

Second, an analysis of the psychometric properties of the questionnaire translated into Spanish was performed. Consequently, the Cronbach’s alpha coefficient was used to estimate the internal consistency of the global questionnaire, considering as optimum values those ranging from 0.70 to 0.95 [25]. We also calculated whether any of the items were deleted in order to check whether all of the items contributed to an optimum alpha coefficient [12]. The correlations between items were calculated with the Pearson correlation coefficient to assess consistency, as well as the item–total scale correlation [26] to prove correlations of each item with the global questionnaire. Adequate internal consistency is considered when the coefficient α is equal to or higher than 0.70 [27]. We interpreted these coefficients, according to Cohen [28], as follows: a low correlation for a coefficient value of 0.1, a moderate correlation for a value of 0.3 and a high correlation for a value of 0.5.

Finally, we performed an analysis of construct validity to ensure that the composition of the experimental scale dimensions corresponded to that of the original scale. Exploratory factor analysis with both the principal components extraction method and the varimax rotation method was performed. These methods are used to identify underlying constructs or factors that can explain the correlations among a set of items and to summarise a large number of items with a smaller number of derived items, termed factors [29,30]. The inclusion criterion for considering a factor as valid was eigenvalues over 1 [31].

Before performing the exploratory factor analysis, we needed to prove its suitability. We did so by using a correlation matrix, the Kaiser–Meyer–Olkin (KMO) measurement of sampling adequacy, and Bartlett’s test of sphericity. The KMO measurement of sampling adequacy tests whether the partial correlations among items are small. A KMO value of greater than 0.7 indicates a strong correlation, meaning that factor analysis should be a useful technique.

Bartlett’s test of sphericity assesses whether the correlation matrix is an identity matrix, indicating that the factor model would be inappropriate [27].

The study was approved by the Experimental Ethics Committee of the University of Malaga (Malaga, Spain). CEUMA Registration No.: 135-2021-H.

## 3. Results

A total of 10 subjects participated in the pilot phase, all of them undergraduate students of the Physiotherapy degree at the University of Seville, who agreed to voluntarily participate in order to identify possible difficulties in understanding the Spanish version. All agreed that it was not necessary to modify anything, so the Spanish version was considered definitive, and the psychometric properties were analysed.

The study sample consisted of 100 participants, who received the online form either directly or through other participants who distributed it. Of these 100 questionnaires obtained, 2 had to be discarded because one of the questions was left blank, so the final sample consisted of 98 completed questionnaires. Sociodemographic data were collected for the entire sample: age, province of residence, employment status and whether they had a diagnosis of ED.

Of the total sample, only one participant reported being diagnosed with ED. The mean age of the participants was 27.86 (SD 10.61); all were residents of Spain, mostly from the north of the country; most of them were also working (*n* = 64), with only 2 older participants who were already retired.

### 3.1. Questionnaire Scores

Table 1 shows the scores obtained for the questionnaire as a whole and for each of its component items. As can be seen, participants obtained a mean score of 22.3 (SD 2.7), implying normal erectile function. However, 23 results of mild dysfunction (*n* = 23) and 2 of mild to moderate dysfunction (5.1%) were identified.

### 3.2. Reliability Analysis

To analyse the questionnaire’s reliability, the internal consistency was evaluated using Cronbach’s alpha (Cortina 1993). This coefficient ranges between 0 and 1, values over 0.60 indicating acceptable reliability, and 0.7 high reliability.

A Cronbach’s alpha coefficient of 0.75 was obtained for the total of the final questionnaire, indicating high reliability. The internal consistency analysis shows that the elimination of the fifth question from the questionnaire increases the reliability (Table 2).

This fifth item is precisely the one that correlates worst with the rest of the items, but the research team decided not to eliminate it in order to respect the original structure of the questionnaire, and because its elimination did not imply a significant increase in the Cronbach’s alpha.

### 3.3. Validity Analysis

To assess construct validity, we performed an exploratory factor analysis with the principal components extraction method. Factor analysis is primarily used to simplify the information from the correlation matrix and to facilitate its interpretation.

To check the suitability of applying factor analysis we calculated the Kaiser–Meyer–Olkin (KMO) statistic, a ratio that compares observed correlation coefficients with the partial correlation coefficients for the set of variables. As can be seen in Table 3, as this statistic has a value of 0.784 (>0.5), factor analysis is considered appropriate.

In Table 3, we check the result of Bartlett’s test of sphericity, which indicates whether the correlation matrix is significantly different from the identity matrix. As the significance level is indeed lower than 0.05, we can reject the null hypothesis that the matrix is equal to the identity matrix and, therefore, there is an acceptable level of correlation between the variables.

In checking the anti-image correlation matrix (Table 4), we observed that the values of the diagonal of this matrix are greater than 0.5. The diagonal of the matrix contains a measure of sampling adequacy for each variable, similar to that obtained in the KMO, but for each variable individually considered.

The eigenvalue (the percentage of the total variance explained by each factor) provides guidance on the number of factors, dimensions or components that make up a questionnaire, and by default those whose eigenvalue is greater than the value of 1 should be selected. Table 5 and Table 6 present the dimensions extracted from the IIEF-5 scale, and the correlation of each of the questionnaire items with the main dimension is identified.

## 4. Discussion

In this paper, we present a cross-cultural adaptation of the IIEF-5 questionnaire, which is highly reliable and valid according to the results obtained in the psychometric analysis. Moreover, it has become the first questionnaire validated in Spanish for the early detection of ED, which we consider highly relevant for clinical practice in clinics where patients with pelvic floor dysfunction are treated.

For the creation of the original IIEF-5 scale, the participants selected were men diagnosed with ED who were participating in clinical trials investigating the safety and efficacy of Viagra (Sildenafil). On the other hand, a control group of men with no history of erectile dysfunction was recruited from volunteers at an outpatient community health centre [15]. The Spanish validation of the Erection Hardness Score (EHS) was carried out with patients referred by general practitioners to the general urology clinic [32]. In our case, we included the general population, regardless of whether or not they had had a diagnosis of ED. This was not a problem, as the aim of the study was to show the reliability and validity of the electronic version of the IIEF-5.

The “gold standard” questionnaire for assessing function in patients with ED is the SHIM questionnaire. However, it is a long questionnaire composed of 15 items, of which only 6 are able to adequately discern between men with and without ED. An advantage of the IIEF-5 is its speed of administration, as it has only 5 items that show good discriminatory ability, with a specificity of 88% and a sensitivity of 98%. In addition, there are other questionnaires that evaluate erectile dysfunction or some related element: Erection Hardness Scale (EHS), Quality of Erection Questionnaire (QEQ), The Erectile Function Pineda Visual Analog Scale (EFP-VAS). Although some authors consider the IIEF-5 scale to be a long questionnaire and difficult to use in daily urological practice [32], our results show that the electronic version in Spanish is an adequate tool for the diagnosis of ED.

This study has not only provided a methodology for cross-cultural adaptation but also validated its online administration, which does not require face-to-face administration in a doctor’s office, which is of particular interest nowadays due to the COVID-19 pandemic. Online administration is therefore a safe, fast, cheap, and simple method of administration, which is of benefit in daily practice. In the Netherlands [33], the SHIM and IIEF-5 scales were validated in their electronic version. They performed the validation on a sample of 179 patients, using “click or click here to type text”, showing an equivalence with the printed paper format. The differences between this study and ours were basically the anonymity of the participants, which prevented us from being able to compare the results of the online version with a printed version. The preservation of anonymity and the sampling method, which allowed the form to be disseminated via Google Forms to a population unknown to the research team, also prevented us from carrying out sensitivity testing by means of a test–retest.

It is important to bear in mind the advantages and disadvantages of online administration, one of which is the possibility of accessing a large number of people at low cost and speed, aspects that are difficult to achieve in surveys administered in person [34]. Moreover, as research by Pratesi et al. [35] confirms, this form of administration allows much easier and quicker access to busy and hard-to-reach people compared to face-to-face or telephone surveys. Furthermore, the accuracy will be much higher in online surveys as participants register their responses through easy-to-choose buttons. Traditional methods require human intervention and, according to studies, this increases the margin of error by 10% [36].

The major disadvantages of self-administered surveys via the Internet are related to the difficulty in locating representative samples due to coverage problems, as not everyone has access to the Internet nowadays [37,38,39,40,41]. It should not be forgotten that the network requires not only the necessity of having equipment at one’s disposal, such as a computer, but also a series of basic skills for its correct use [36], something that is still lacking in some parts of the population.

Knowing that the IIEF-5 scale was to be administered online, the format of the questionnaire was suitable for this purpose and was not expected to present serious difficulties for the participants. In addition, a more extensive or complex questionnaire in its entirety would not be administered via the Internet in such a satisfactory way.

In relation to the psychometric analysis, our Spanish version of the IIEF-5 presents a Cronbach’s alpha of 0.761, which means that it has high reliability. In the internal consistency analysis, we found strong correlations between the different items in relation to the total scale (*p* < 0.005), which indicates that the questionnaire is coherent. Only item 5 presents a moderate-weak correlation coefficient with respect to the rest of the scale, and its elimination could mean an increase in the scale’s reliability coefficient, but the authors decided not to eliminate it because the increase in the Cronbach’s alpha was not very significant (see Table 2) and because they did not want to modify the original content of the scale, taking into account the criteria and recommendations of the committee of experts who considered this to be the case. Furthermore, according to Shaw and Young [42], there is no obligation to eliminate items with an item—total correlation coefficient between 0.2 and 0.3, considering them valid.

The reliability of this Spanish version of the IIEF-5 is high, as is the reliability of the long version and of the original versions. Unfortunately, we have not been able to compare it to Pineda’s scale [12], written in Spanish, because that study lacked a psychometric analysis.

As for the construct validity of the questionnaire, our factor analysis obtained a single principal component, which explains more than 50% of the variance, and which correlates strongly with all the items of the scale, with the exception of item 5, which has a weaker correlation. This finding makes us consider the soundness of the instrument and how well focused it is for the detection of ED in a specific way, with the certainty that it is assessing what it is intended to assess quickly and easily. For the intended purpose of using this questionnaire, we consider it an important advantage, since it will allow us to refer to the urologist with fewer doubts about the results of the questionnaire.

We believe that this cross-cultural adaptation has important implications for the daily clinical practice of health professionals working with patients with pelvic floor dysfunction to enable the detection of ED. Its adaptation into Spanish was a pending issue, since its use is widespread, and to date, versions with different translations are being used indistinctly in some health centres without the necessary scientific validation.

Therefore, we suggest the use of this questionnaire for use in the Spanish-speaking population and encourage professionals who need it to use it to facilitate the work in multidisciplinary teams and improve the detection and subsequent treatment of this dysfunction, which is not always reported by affected patients.

The main limitation of this study has been the impossibility of analysing criterion validity or concurrent validity, as no validated item has been found in Spanish. The only scale found in Spanish was the EFP-VAS. In this study carried out by Pineda [12] in the Mexican population, it should be noted that the study sample selected was mainly Spanish-speaking, but there is no assurance that this questionnaire has been validated in the entire Spanish-speaking population. Moreover, there is no record of the results of the psychometric analysis [12].

Another limitation of the study was the sample collection. Since the questionnaire was administered anonymously and because of the way in which it was distributed, it was not possible to carry out a test–retest, making it impossible for the participants to answer the questionnaire again, thus limiting the possibility of reevaluating the data obtained from each of them.

For all these reasons, we believe that in future studies, it would be advisable to administer the questionnaire in a clinical context, respecting anonymity at all times, but with the generation of codes that allow the questionnaires to be matched in order to correct this limitation and to be able to carry out a test-retest, which would also imply a different form of sampling to that used in this study, thus improving reliability.

## 5. Conclusions

The results of the present study demonstrate that the IIEF-5 questionnaire is adequately reliable and valid, and easy to administer and disseminate in an online format. This Spanish version could help identify patients with ED and refer them to the physician (in charge of the medical diagnosis). Moreover, it is a useful tool for the physiotherapist to measure the evolution of the treatment in these cases. Moreover, it is a questionnaire that is easy to fill in and disseminate via an online platform, which is also a great advantage for the patient.

## Figures and Tables

**Table 1 ijerph-19-03115-t001:** Result of the scores obtained in the Spanish version of the IIEF-5.

*N* = 98	Min	Max	Average	Standard Deviation
**Item 1**	2	5	4.33	0.729
**Item 2**	0	5	4.59	0.823
**Item 3**	1	5	4.38	0.947
**Item 4**	2	5	4.65	0.660
**Item 5**	2	5	4.36	0.692
**Total IIEF-5 score**	12.00	25.00	22.3061	2.75609

**Table 2 ijerph-19-03115-t002:** Table of item—total scale correlations.

	Scale Mean if Item Was Deleted	Scale Variance If Item Was Deleted	Corrected Item—Total Correlation	Squared Multiple Correlation	Cronbach’s Alpha Value If Item Was Deleted
**Item1**	17.98	5.113	0.592	0.386	0.684
**Item 2**	17.71	4.990	0.525	0.291	0.707
**Item 3**	17.93	4.356	0.593	0.376	0.683
**Item 4**	17.65	5.280	0.620	0.452	0.681
**Item 5**	17.95	6.070	0.307	0.115	0.775

**Table 3 ijerph-19-03115-t003:** Kaiser–Meyer–Olkin coefficient and Bartlett’s test of sphericity.

**Kaiser–Meyer–Olkin Index**		0.784
**Bartlett’s** **Test of Sphericity**	Approx. Chi-square	117.154
	df	10
	Sig.	0.000

**Table 4 ijerph-19-03115-t004:** Anti-image matrix with sampling adequacy measure (^a^ Sampling adequacy measures).

**Item 1**	0.792 ^a^	−0.105	−0.219	−0.356	−0.136
**Item 2**	−0.105	0.827 ^a^	−0.142	−0.272	−0.156
**Item 3**	−0.219	−0.142	0.805 ^a^	−0.315	−0.141
**Item 4**	−0.356	−0.272	−0.315	0.737 ^a^	0.082
**Item 5**	−0.136	−0.156	−0.141	0.082	0.766 ^a^

**Table 5 ijerph-19-03115-t005:** Principal component extraction method.

Component	Initial Eigenvalues		
	Total	% of Variance	% Accumulated
**Item 1**	2.582	51.647	51.647
**Item 2**	0.886	17.725	69.371
**Item 3**	0.625	12.494	81.865
**Item 4**	0.509	10.176	92.041
**Item 5**	0.398	7.959	100.000

**Table 6 ijerph-19-03115-t006:** Analysis of the extracted principal component.

	Component 1
**Item 1**	0.780
**Item 2**	0.715
**Item 3**	0.780
**Item 4**	0.805
**Item 5**	0.455

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
