# Peer review of "Spanish Cross-Cultural Validation of the Electronic Version of the International Index of Erectile Function-5 (IIEF-5)"

_ijerph, 2022, doi:10.3390/ijerph19053115_

Round 1
Reviewer 1 Report
Thank you for giving me to review your manuscript. This manuscript is interesting and meaningful for considering male sexual issues for their better health. Regarding the contents, the following revision should be considered for the quality of research.
The title should be more specific regarding study design.
Line 67-79
This paragraph should be divided into two paragraphs, importance of the instruments and examples of the instruments.
The introduction should clearly include this study's research question. There are many studies regarding this research topic, especially in different countries.
In the sample section of the method, there are no descriptions regarding sample calculation. Therefore, the authors should descript the sample size calculation.
The statistical analysis should be more described. The authors should explain how to deal with each variable, referring to previous studies.
The discussion should describe the limitation of sampling bias and the results' applicability to other settings, and the future investigation in the limitation part, especially regarding the analysis.
Author Response
The authors are very grateful for your assessement and suggestions to improve the manuscript.
1.-Line 67-79: This paragraph should be divided into two paragraphs, importance of the instruments and examples of the instruments.
We have divided the paragragh into two, as you have suggested.
2.-The introduction should clearly include this study's research question. There are many studies regarding this research topic, especially in different countries.
We have included the study´s research question al the end of the introduction, just before the main objective.
3.- In the sample section of the method, there are no descriptions regarding sample calculation. Therefore, the authors should descript the sample size calculation.
We have extended the explanation for the estimation of the simple size.
4.- The statistical analysis should be more described. The authors should explain how to deal with each variable, referring to previous studies.
We have provided a larger explanation and references.
5.- The discussion should describe the limitation of sampling bias and the results' applicability to other settings, and the future investigation in the limitation part, especially regarding the analysis.
A new paragraph has been added in the discussion section to adress the issues concerning the limitations.
6.- The authors have detected another error which has been solved by removing the two last paragraphs of the Statistical Analysis section. We are working on another project related to another scale validation, and we had included a text that did not correspond to this study. We apologize for this mistake.
Reviewer 2 Report
Thank you for the opportunity to review a manuscript entitled „Spanish transcultural validation of the electronic version of the Sexual Health Inventory for Men (SHIM)” and its evaluation for possible publication in the International Journal of Environmental Research and Public Health.
The topic is interesting and worth attention. The study is well designed and presented. The introduction is a bit difficult to follow and needs improvement. Also, some minor issues should be addressed before publication.
I suggest using the term “cross-cultural validation” instead of “transcultural validation”. See definitions in Huang W.Y., Wong S.H. (2014) Cross-Cultural Validation. In: Michalos A.C. (eds) Encyclopedia of Quality of Life and Well-Being Research. Springer, Dordrecht.
Abstract: No need to use numbered points. Please, spell out IIEF-5. Do not use inverted commas for Sexual Health Inventory for Men. In line 27, please use an abbreviation for Sexual Health Inventory for Men as before.
Generally, about the introduction - it is a bit too long. What is missing here is an explanation and proper naming of the scale studied. In the literature, the terms SHIM and IIEF-5 are used separately. Most researchers use only one term. Can you please explain why there are two terms given for the same scale?
Specifically, a paragraph starting with line 52 would bring additional value by adding that the same happens with specialists. Even if the patient is treated by specialists, such as cardiologists who see many more patients with ED due to sharing common pathophysiology between CVD and ED, still problems of sexual functioning are not addressed. This is in contract with patients’ expectations. This topic is presented in the paper entitled “Sexual health of male cardiac patients – present status and expectations of patients with coronary heart disease” by KaÅ‚ka et al.
The paragraph starting with line 62 is short and is a bit outside of the topic. It can be therefore removed, particularly because there is not reference given to support this.
Line 76, please give the term in full before using an abbreviation – relates to SHIM.
Language should be improved. There are a lot of minor grammatical errors and typos. For example, line 20 should be ten individuals; line 41 should be suffer from; line 52 abbreviation should be used for erectile dysfunction. But also some common words are used inappropriately. Maybe you can consider a review by a language speaker.
Methods. There was a lot of done by translators and experts in this study. Are they the authors of this paper? If yes please put their initials in the methods next to tasks that were conducted by them. If not, I suggest mentioning them in the Acknowledgments.
Results. Table 3 and Table 5 seem to be in Spanish. Please correct.
General. Can you please add a Spanish version of the questionnaire into the appendix so that it can be used by other researchers?
Author Response
The authors are very thankful for your assessemnt and suggestions to improve the manuscript.
1.- I suggest using the term “cross-cultural validation” instead of “transcultural validation”. See definitions in Huang W.Y., Wong S.H. (2014) Cross-Cultural Validation. In: Michalos A.C. (eds) Encyclopedia of Quality of Life and Well-Being Research. Springer, Dordrecht.
We totally agree with you, and the title has been modified.
2.- Abstract: No need to use numbered points.
The numbers have been removed.
3.- Please, spell out IIEF-5.
It has been spelled out.
4.- Do not use inverted commas for Sexual Health Inventory for Men.
The inverted commas have been removed.
5.-In line 27, please use an abbreviation for Sexual Health Inventory for Men as before.
We changed the term for IIEF-5.
6.- Generally, about the introduction - it is a bit too long. What is missing here is an explanation and proper naming of the scale studied. In the literature, the terms SHIM and IIEF-5 are used separately. Most researchers use only one term. Can you please explain why there are two terms given for the same scale?
The authors have dealt this issue several times, and after reading your comments, we have decided to use the term IIEF-5 for the short form of the SHI scale, since this shirt form is the one to be validated in Spanish.
7.- Specifically, a paragraph starting with line 52 would bring additional value by adding that the same happens with specialists. Even if the patient is treated by specialists, such as cardiologists who see many more patients with ED due to sharing common pathophysiology between CVD and ED, still problems of sexual functioning are not addressed. This is in contract with patients’ expectations. This topic is presented in the paper entitled “Sexual health of male cardiac patients – present status and expectations of patients with coronary heart disease” by KaÅ‚ka et al.
Thank you very much for providing a reference. Th eparagraph has been changed and now reflects the fact that other medical specialists also cope with patients with erectile dysfunction.
8.- The paragraph starting with line 62 is short and is a bit outside of the topic. It can be therefore removed, particularly because there is not reference given to support this.
The paragraph has been removed.
9.- Line 76, please give the term in full before using an abbreviation – relates to SHIM.
The full name of the scale has been indicated.
10.- Language should be improved. There are a lot of minor grammatical errors and typos. For example, line 20 should be ten individuals; line 41 should be suffer from; line 52 abbreviation should be used for erectile dysfunction. But also some common words are used inappropriately. Maybe you can consider a review by a language speaker.
We have checked the manuscript and rectified the mistakes. Nevertheless, the document has been sent and reviewed by professionals in translation and edition of sceintifc literatura.
11.- Methods. There was a lot of done by translators and experts in this study. Are they the authors of this paper? If yes please put their initials in the methods next to tasks that were conducted by them. If not, I suggest mentioning them in the Acknowledgments.
Some authors did the translation into Spanish, their initials have been included in Methods. The back-translation was carried out by other experts, and are now cited in Acknowledgements.
12.- Results. Table 3 and Table 5 seem to be in Spanish. Please correct.
Tables corrected, we apologize for that mistake.
13.- General. Can you please add a Spanish version of the questionnaire into the appendix so that it can be used by other researchers?
We have now added the Spanish version of the IIEF-5 as a supplementary file.
14.- The authors have detected another error which has been solved by removing the two last paragraphs of the Statistical Analysis section. We are working on another project related to another scale validation, and we had included a text that did not correspond to this study. We apologize for this mistake.

Round 2
Reviewer 1 Report
The manuscript has been considerably improved. I think that this paper is suited for inclusion in our journal.